# Estimation of Fatigue Crack AE Emissivity Based on the Palmer–Heald Model

**Vera Barat** [1,2]**, Artem Marchenkov** [1,]*** and Sergey Elizarov** [2]

[1]    Moscow Power Engineering Institute: 14, Krasnokazarmennaya str., 111250 Moscow, Russia;
       vera.barat@mail.ru
[2]    LLC "Interunis-IT": 20b, Entuziastov sh., 111024 Moscow, Russia; serg@interunis-it.ru
*    Correspondence: art-marchenkov@yandex.ru

**Abstract:** This article is devoted to materials testing by the acoustic emission (AE) method, which is the analysis of models and diagnostic parameters to assess the probability of detection of a defect in steel structures. The paper proposes to evaluate the emissivity of the material quantitatively by the number and dynamics of the accumulation of acoustic emission impulses. Experimental studies were carried out on pearlitic structural steels, including the loading of samples with fatigue cracks. It was established that the number of AE impulses emitted during loading of an object with a fatigue crack is a random variable corresponding to the normal distribution law. The results show that an estimate of the number of AE impulses emitted during the loading of samples with fatigue cracks can be obtained by distributing the multiplicative parameter $D$ of the Palmer-Heald model by taking into account the maximum value of the applied load.

**Keywords:** acoustic emission; diagnostics; fatigue crack; structural steel; emissivity

## 1. Introduction

The acoustic emission (AE) method is successfully used to detect defects in items and structures for various purposes. Fatigue crack is one of the most widespread and dangerous defects which could be successfully detected with help of the AE method. One of the most complex and relevant directions in the development of the AE-diagnostic methodology is the establishment of a relationship between the parameters of a fatigue crack and the AE impulses parameters [1].

The strong relationship between the microstructure of materials and AE parameters complicates the construction of a fatigue crack propagation model. In the theory of AE diagnostics, a number of analytical and phenomenological models have been developed that connect the crack parameters and AE data during the static loading, e.g., the Dunegan model [2], the Palmer-Heald model [3], and the Holt and Goddard model [4]. In the case of propagation of a fatigue crack under cyclic loading, an equation similar to the Paris–Erdogan law has been proposed [5,6]. The disadvantage of existing models is their parametric uncertainty—the functional relationship between the mechanical characteristics of the material and the AE parameters is established with accuracy to coefficients depending on the material properties, structure features and operating conditions. The parametric uncertainty of the models does not allow a quantitative estimation of the parameters of AE impulses induced by the crack.

The active development of technology for scanning electron and laser confocal microscopy [7] and the improvement of testing and measuring equipment contribute to the development of a phenomenological approach in acoustic emission. Previous studies [8–10] included analyses of AE data during fatigue crack propagation with identification of the fatigue crack propagation stage and classification of various AE sources that appeared during the crack growth. The disadvantage of an

empirical approach is the difficulty of data summarizing and processing as well as the need, in many experiments, to substantiate and confirm the empirical relationships. The complex analytical and phenomenological approach to the fracture process, which was proposed in [11], allowed the design of a theoretically confirmed fracture process model which allows the identification of various stages of material fracture based on the parameters of AE impulses.

This study represents a study of the AE signature of fatigue crack, focused on the improvement of industrial applications of the AE testing method. The objective of this study was to assess the emissivity of structural steels during the development of fatigue cracks in order to clarify the probability of fatigue defect detection during acoustic emission testing.

Acoustic emission testing is the only nondestructive testing method for which the assessment of the reliability of defect detection is not implemented in the field practice. Currently, there are only a few papers devoted to evaluation of the reliability of AE testing [12–14]. To assess the reliability of defect detection during AE monitoring, a quantitative diagnostic model should be formed that allows the determination of not just the functional dependence, but also the quantitative values of AE parameters induced by a defect of a certain type in response to a specific loading effect with a certain probability.

One of the main parameters influencing the probability of defect detection by the AE method is emissivity, the integral characteristic of which is $N_\Sigma$—the number of impulses emitted during the loading of the sample. The uncertainty of the quantity of AE impulses is one of the main reasons of defect detection probability evaluation complexity. To assess the probability of defect detection by the AE method, it is necessary to specify the function of the AE source, including the estimation of emissivity, the estimation of the law of distribution of amplitudes and also the function of the acoustic waveguide. There are several fundamental research papers describing AE impulse amplitude distribution laws [15–17] while the possibility of finite element analysis makes it possible to acoustic waveguide precise calculation [14,18,19]. At the same time, an evaluation of the material emissivity and estimation of the number of AE impulses emitted by a defect is a problem that has not been completely solved. At present, emissivity is evaluated either qualitatively (materials are characterized by low, average or high emissivity), or approximately—by one characteristic value [20]. In a number of studies, two methods of non-destructive testing were used together to interpret and predict the $N_\Sigma$ parameter. For example, the joint use of AE and magnetic testing was described in [21] and the joint use of AE and thermography was performed in [22]. The AE parameters and in particular, $N_\Sigma$, are interpreted based on the known diagnostic models of magnetic testing and thermography, respectively.

In this paper, to establish a relationship between the defect parameters and AE data, it is proposed to study the nature of the $N_\Sigma$ change with a stress increase during the sample loading. To quantify parameter $N_\Sigma$ and the nature of its change with an increasing load, it is proposed to use the coefficients of analytical Dunegan [2] and Palmer-Heald [3] models.

Based on the assumption that the number of AE impulses is proportional to the volume of the plastic deformation zone of the material at the crack tip and that the accumulation rate of AE pulses is proportional to the rate of increase of the plastic deformation zone under static loading of the object, the Dunegan model can be written in the form

$$N_\Sigma = AP^n \tag{1}$$

where $P$ is the applied load; $n$ is a power indicator characterizing the stress intensity factor at the crack tip; $A$ is a constant.

The power indicator $n$ determines the form of the dependence $N_\Sigma(P)$ and $A$ is a normalizing coefficient that ensures the correspondence of the quantitative indicators of the model and the experiment.

According to the Palmer–Heald model, the number of AE impulses can be expressed as

$$N_\Sigma = D{\cdot}a{\cdot}\left(sec\left(\frac{\pi}{2}\frac{\sigma}{\sigma_y}\right) - 1\right) \tag{2}$$

where $\sigma$ is the actual stress; $\sigma_y$ is theyield strength; $a$ is half the length of the crack; $D$ is thedimensional coefficient of the model (1/m), depending on the characteristics of the material, temperature and type of stress-strain state.

The number of AE impulses recorded during loading of an object with a crack is determined by the material properties, defect characteristics and loading conditions. In the Dunegan model, the loading parameter is taken into account as the basis of the power-law dependence $P$, the defect characteristic is taken into account using the exponent $n$, and the coefficient $A$ is determined by the material properties. In the Palmer–Heald model, the loading conditions are determined by the $\sigma/\sigma_y$ ratio, the defect characteristics are determined by a coefficient $a$ equal to half the crack length and the value of the coefficient $D$ depends on the material properties.

Thus, the coefficient $D$ of the Palmer–Heald model and coefficient $A$ of the Dunegan model can be considered as parameters that quantitatively characterize the material emissivity (as parameter that closely correlated with AE emissivity), independent of loading conditions, crack length, and stress intensity factor at its apex. This paper describes an approach to the evaluation of the quantity of AE impulses emitted of fatigue crack. The evaluation is based on of the Palmer–Heald model multiplicative parameter $D$, which can be interpreted as an indirect evaluation of AE emissivity of the steel during the crack propagation.

Since the purpose of this study was to estimate the value of the parameter $N_\Sigma$, it is advisable to consider, in the framework of the Palmer–Heald model, an additional parameter $D_a$ equal to the product of $D$ by $a$:

$$D_a = D \cdot a \tag{3}$$

Since the crack length is unknown during industrial AE monitoring, parameter $D_a$ can be used to determine the number of impulses $N_\Sigma$ at a known value of the maximum mechanical stress. Due to the influence of the material microstructure on the values of AE parameters, it is assumed that the estimated parameters will be random and evaluated using a probability distribution. This assumption, as well as the results obtained, is consistent with the Bayesian statistical approach proposed by Vinogradov in [23].

## 2. Materials and Methods

An experimental study was conducted in the framework of which the loading of samples was carried out using the AE method.

For research, three grades of steel were selected—09G2S, 45 and 65G (marked in accordance with Russian state standards). The chemical composition of the studied steels is presented in Table 1.

**Table 1.** The chemical composition of the studied steels.

| Steel | The Content of Chemical Elements (% wt.) | | | | | | | | |
|---|---|---|---|---|---|---|---|---|---|
| | C | Mn | Si | Cr | Ni | Cu | As | S | P |
| 09G2S | ≤0.12 | 1.3–1.7 | 0.5–0.8 | ≤0.3 | ≤0.3 | ≤0.3 | ≤0.08 | ≤0.04 | ≤0.035 |
| 45 | 0.42–0.5 | 0.5–0.8 | 0.17–0.37 | ≤0.25 | ≤0.25 | ≤0.25 | ≤0.08 | ≤0.04 | ≤0.035 |
| 65G | 0.62–0.7 | 0.9–1.2 | 0.17–0.37 | ≤0.25 | ≤0.25 | ≤0.2 | - | ≤0.035 | ≤0.035 |

Each of these steels is widely used in various industries [24]. These steels were investigated in the form of hot-rolled sheets with a thickness of 3...5 mm. Steel sheets were chosen so that each of these steels had a ferrite-pearlite structure, shown in Figure 1.

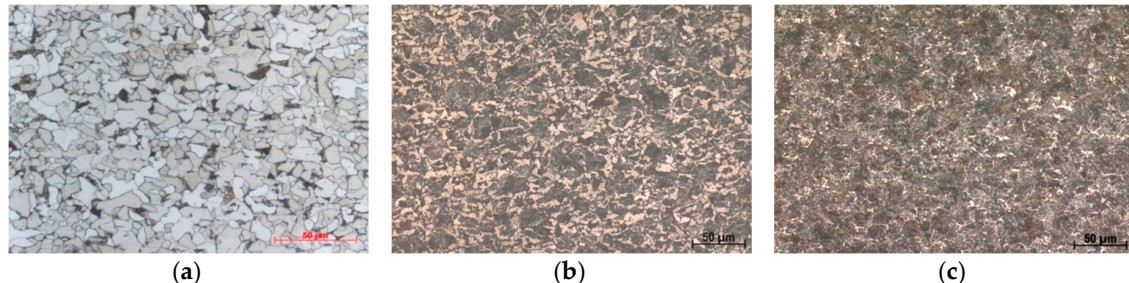

**Figure 1.** Microstructure of the studied steels: (**a**) 09G2S; (**b**) 45; (**c**) 65G.

To assess the actual mechanical properties of the steels under study, static tensile tests of flat samples cut along and across the direction of rolling were carried out in accordance with ISO 6892. Stress–strain curves of the studied steels and their mechanical properties are presented in Figure 2 and Table 2.

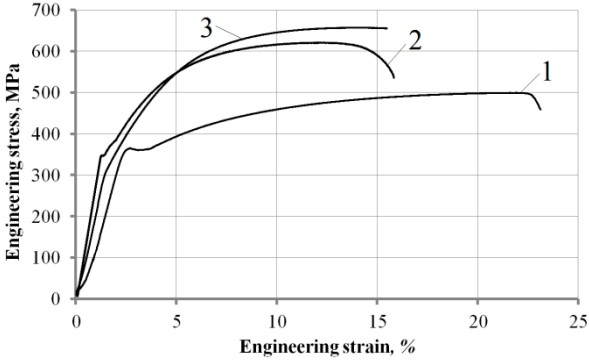

**Figure 2.** Stress–strain curves for the studied steels: 1—09G2S; 2—45; 3—65G.

**Table 2.** Mechanical properties of the studied steels (average values).

| Steel | Yield Strength $\sigma_y$, MPa | Ultimate Tensile Stress $\sigma_u$, MPa | Total Elongation $A$, % |
|---|---|---|---|
| 09G2S | 365 | 503 | 23 |
| 45 | 333 | 622 | 15 |
| 65G | 325 | 654 | 15 |

The studied steels were tested for static tension using a loading scheme adopted during industrial AE testing [25]. For this, samples were made from each of steels; the sample design is shown in Figure 3.

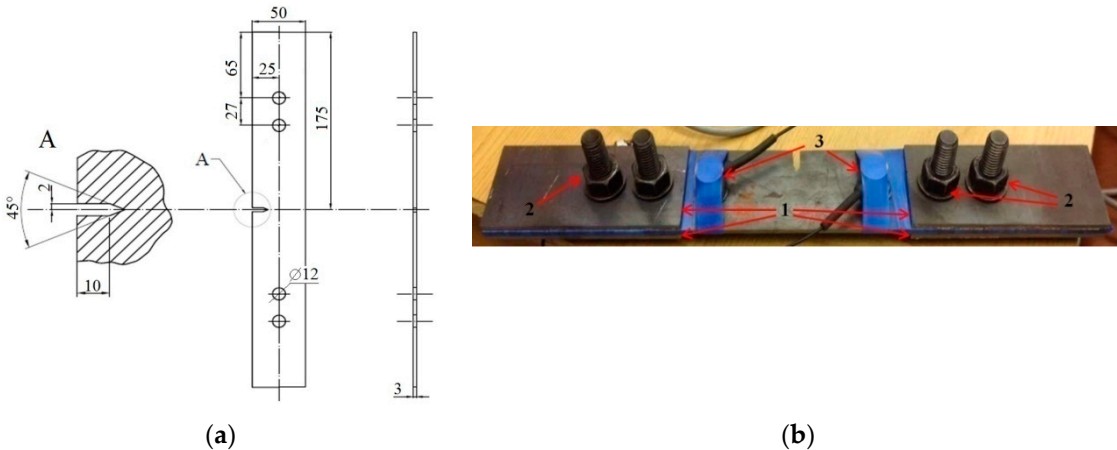

**Figure 3.** The design of the test sample (**a**) and a photograph of one of the samples (**b**): 1—damping pads, 2—fixing bolts, 3—acoustic emission sensor.

Flat samples with a working part width of 50 mm with an edge notch were made taking into account the recommendations of Russian State standard GOST 25.506-85, which describes the determination of crack resistance characteristics of structural materials. The notch width was chosen as equal to 4 mm, the opening angle $\Theta = 45°$, and the notch depth $h = 10$ mm. Holes with a diameter of 12 mm are provided for fixing the damping pads, which reduce the level of acoustic interference of the loading machine. In total, 50 samples were made of 09G2S steel (of which 30 samples were made of sheets of a 3-mm thickness and 20 samples were made of sheets of a 5-mm thickness), 10 samples were made of 4 mm sheets of 45 steel, and 10 samples were made of 4 mm sheets of 65G steel.

A fatigue crack was grown on each of the test samples. For this, the sample was subjected to cyclic tensile loading along the pulsation cycle (cycle asymmetry coefficient $R_c = 0$) with a frequency of 5 Hz and a maximum cycle load of $\sigma_{max} \approx 0.6 \cdot \sigma_y$. The $\sigma_{max}$ values amounted to 210 MPa for samples of steel 09G2S and 190 MPa for steels 45 and 65G. The number of loading cycles was chosen so that the fatigue crack developing under cyclic loading in the lateral notch region would reach a certain length. Nominal crack lengths ranged from 3 to 15 mm. Thus, the total length of the notch and crack was approximately 15...28 mm, depending on the specimen.

Before the main tests, an experimental study of the stress–strain state of the metal at the tip of the crack was carried out on the test specimens made of 09G2S steel with a pre-grown fatigue crack. Studies were performed using the indentation method. For this, indents were applied on the side surface of the sample with a Vickers pyramid under a load of 0.015 kg and the microhardness values *HV0.015* were determined. The indents were applied on an Instron Tukon 2500 Vickers hardness tester in rows with a pitch between the indentions of 50 μm and the distance between the rows of 50 μm (Figure 4). The values of the hardness *HV0.015* of the metal at each point assessed the presence of plastically deformed metal. Thus, an increase in the *HV0.015* hardness in the defined point relative to the hardness of the undeformed metal indicates its plastic deformation, and the higher the metal hardness value, the greater the degree of its plastic deformation [26].

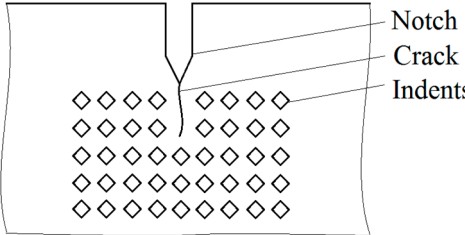

**Figure 4.** The methodology for the study of the deformed volume in the notch zone on samples of 09G2S steel.

Testing of specimens with fatigue cracks was carried out on an Instron 8801 machine. Acoustic signals were recorded during the tests using the A-Line 32D industrial system. The measuring path consisted of resonant transducers GT200 (LLC "Global test") with a resonance frequency of 165 kHz and preamplifiers PAEF-014. The peak noise of the preamplifier amounted to 26 dB in reference to 0 dB at 1 μV. The peak acoustic noise level after installing the sample in the grips of the testing machine was 34 dB. In accordance with the recommendations [25], the threshold for discrimination of acoustic signals was set to 40 dB. A diagram of the loading cycle of the sample is presented in Figure 5.

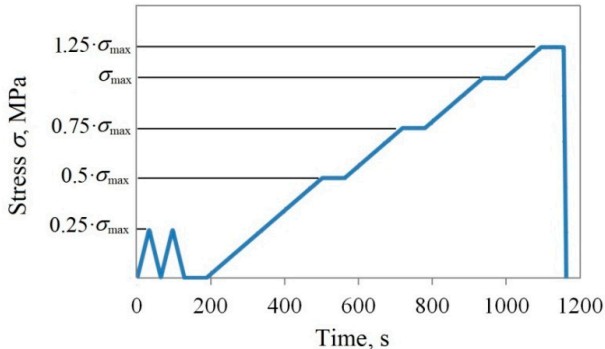

**Figure 5.** Diagram of the loading cycle.

Two triangular cycles at the first stage of loading are necessary to equalize the internal mechanical stresses that occurred during the manufacture of samples and when placing them in equipment. The acoustic activity that occurs during such loading is usually not associated with the fracture processes. The main loading stage consists of four main stages: in the first, the load gradually increases from 0% to 50% $\sigma_{max}$, in the second, from 50% to 75% $\sigma_{max}$, and in the third, from 75% to 100% $\sigma_{max}$. The fourth stage is carried out with a stress excess of $\sigma_{max}$ for 25%. Between the areas of stress increase, a load is held for 90 s. For industrial AE testing, exposure periods are necessary to make a decision on the termination of loading in the event of an emergency. When conducting experiments in laboratory conditions, stress holding sections are introduced to ensure compliance with industrial testing conditions.

In each experiment, the length of the fatigue crack was measured before and after the tests. During loading, parametric information was recorded—the value of the loading force and the position of the lower grip of the machine, the parameters of the AE impulses (amplitude, energy, duration and rise time), as well as the primary information—waveforms of AE impulses.

## 3. Results

According to the results of the tests for each sample, two dependences were obtained—the number of AE impulses from the $\sigma/\sigma_y$ ratio and from the applied load $P$, which were used to analyze the emissivity of the material according to the Palmer–Heald and Dunegan models. The parameters of the analytical models were estimated using nonlinear approximation of empirical data by a power function and a secant function.

The dependence $N_\Sigma(P)$, the number of AE impulses on the applied load, was used as the initial data for the Dunegan model and the coefficients of the model $A$ and $n$ were determined by minimizing the objective function by the gradient method. In the Palmer–Heald model, only one parameter is determined—$D_a$—and the argument of the secant function was set based on the value of $\sigma_y$ determined during mechanical tests. The actual stress $\sigma$ was determined as the quotient of the load and the residual area of the specimen under the assumption of uniformly accelerated crack growth. Examples of the obtained experimental dependences and their approximations for each steel are presented in Figure 6.

The results of approximation of all 70 sets of experimental data obtained for various values of the lengths of fatigue cracks show a high degree of agreement with the theoretical models of Dunegan and Palmer–Heald.

For each of the three studied parameters—$A$, $n$, and $D_a$—the errors and the confidence interval of estimation were determined. The average errors in estimating were the following: for parameter $D_a$ it was 2.3%, for parameter $n$ it was 6.7%, and for parameter $A$ it was ~70%. Due to the large error in determining parameter $A$, which can be explained by the poor conditionality of the calculations, coefficient $D_a$ of the Palmer–Heald model was used to estimate parameter $N_\Sigma$.

The power-law indicator $n$ of the Dunegan model was calculated to characterize the degree of danger of a crack and to verify the locally dynamic criterion for classifying AE sources in accordance with [25].

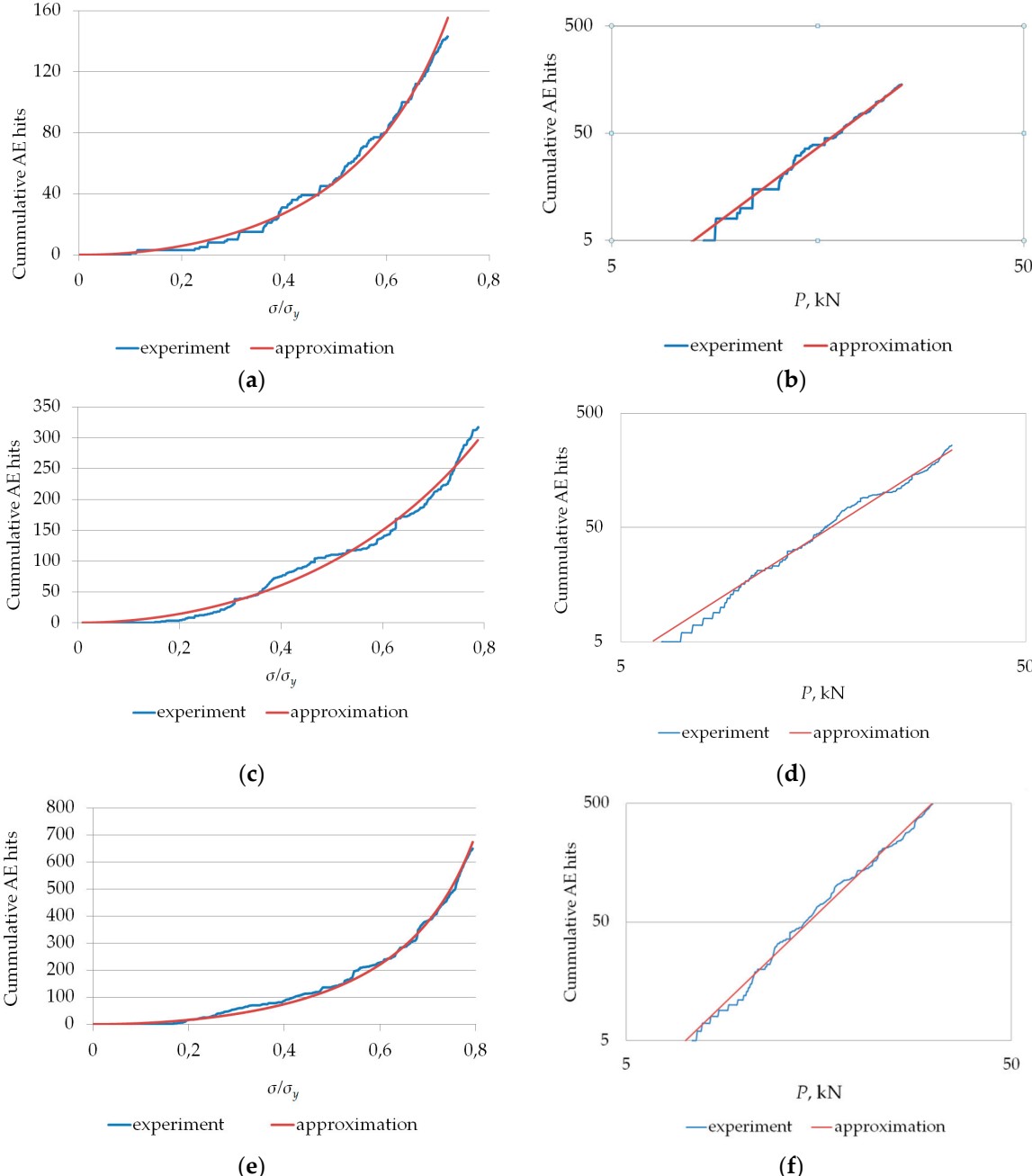

**Figure 6.** The number of AE impulses $N_\Sigma$ depending on the current stress *(σ/σ$_y$)* and load *P*: (**a**,**b**) steel 09G2S, the initial length of the fatigue crack is 18.45 mm; (**c**,**d**) steel 45, the initial length of the fatigue crack is 20.80 mm; (**e**,**f**) steel 65G, the initial length of the fatigue crack is 17.81 mm.

For each sample, the $D_a$ and $D$ parameters were calculated in the Palmer–Heald model. Figure 7 shows, as an example, the empirical distribution of parameters $D_a$ and $D$ for samples of 3-mm thick of 09G2S steel. Visually, the distributions shown in Figure 7 comply with the normal law and to verify this hypothesis, statistical tests of fit for distributions were used. The results are presented in Table 3.

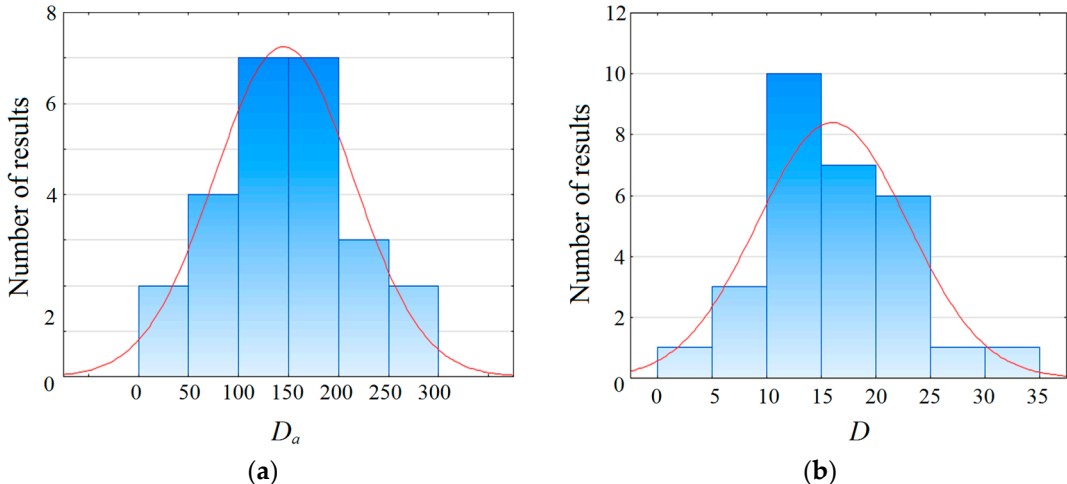

**Figure 7.** Empirical distributions of parameter $D_a$ (**a**) and parameter $D$ (**b**) included in the Palmer–Heald model.

**Table 3.** Statistical tests of fit for distributions results.

| Distribution | Anderson-Darling Test | | Pearson $\chi^2$ Test | |
|---|---|---|---|---|
| | $A^2$ | $p$-Value | $\chi^2$ | $p$-Value |
| **D-Parameter** | | | | |
| Normal | 0.21 | 0.98 | 1.12 | 0.57 |
| Log-Normal | 0.63 | 0.61 | 3.12 | 0.21 |
| Weibull | 0.31 | 0.93 | 1.78 | 0.41 |
| **$D_a$-parameter** | | | | |
| Normal | 0.25 | 0.96 | 1.88 | 0.39 |
| Log-Normal | 0.49 | 0.75 | 2.62 | 0.27 |
| Weibull | 0.56 | 0.68 | 3.32 | 0.19 |

For the fit tests, two criteria were chosen: the Pearson $\chi^2$ criterion, which is the most widespread for determining compliance with a certain type of distribution law and the non-parametric Anderson–Darling test, which retains the reliability in the case of limited sample sizes. The values of the test statistics for the normal distribution for the log-normal distribution and the Weibull distribution law are presented in Table 3. The $A^2$ value corresponds to the Anderson–Darling test and the $\chi^2$ corresponds to the Pearson test. These test statistic values are considered together with probability $p$-values. Since the $p$-values exceed 0.05, all the considered distributions can be used to represent the experimental data. However, the smallest values of the test statistics corresponding to the least deviation of empirical data from the theoretical law, and the largest $p$-values characterizing the greater reliability of the results are observed for the normal distribution law. Table 4 shows the mean values and standard deviations (STD) of the $D$ and $D_a$ parameters for samples with a thickness of 3 mm and 5 mm. The ratio of the same type parameters corresponding to different sample thicknesses turned out to be approximately equal to the thickness ratio (0.6).

**Table 4.** Parameters of $D$ and $D_a$ distributions for various thickness.

| Thickness, mm | $D_a$ | | $D$, 1/mm | |
|---|---|---|---|---|
| | Average Value | STD | Average Value | STD |
| 3 | 144.1 | 68.8 | 15.9 | 6.8 |
| 5 | 256.9 | 140.54 | 26.8 | 14.30 |

The average values and standard deviations of the parameters $D$ and $D_a$ for all the steel grades are shown in Table 5. Given the linear dependence of the distribution of $D$ and $D_a$ parameters on the thickness of the sample, for the case of a metal stress–strain state constancy, the average values and standard deviations (STD) can be estimated as a function of sample thickness $t$.

**Table 5.** Parameters of $D$ and $D_a$ distributions for steels under study.

| Steel | $D_a$ | | $D$, 1/mm | |
|---|---|---|---|---|
| | **Average Value** | **STD** | **Average Value** | **STD** |
| 09G2S | 51 t | 25 t | 5.3 t | 2.6 t |
| 45 | 107 t | 35.7 t | 7.8 t | 3.1 t |
| 65G | 289 t | 64.8 t | 15.3 t | 4.2 t |

The power exponent $n$ of the Dunegan model has a more deterministic character. Its value correlates with both the crack length and the stress intensity factor at the crack tip. The dependence of the exponent $n$ on the crack length is shown in Figure 8.

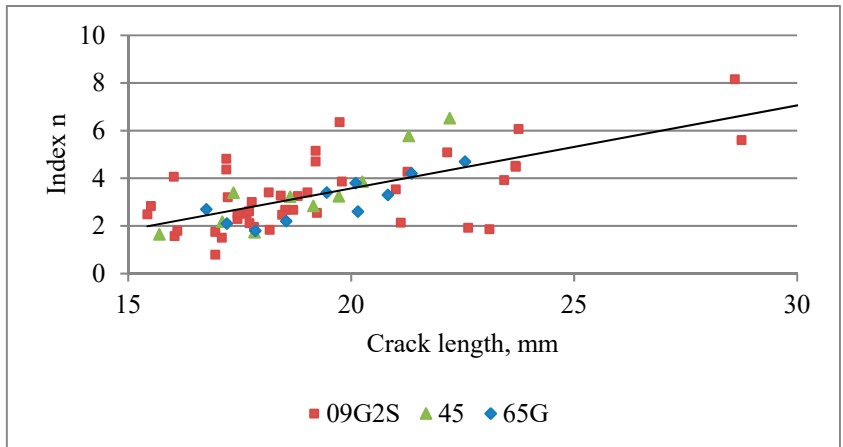

**Figure 8.** Dependence of the power exponent $n$ of the Dunegan model on the crack length.

Despite the high value of the correlation coefficient $r$ of the length of the fatigue crack and the parameter $n$ $(r = 0.85)$ for small sizes of the defect, there is a significant scatter in the exponent $n$, which can be explained by the inhomogeneous microstructure of the material, as well as the difference in the nature of crack propagation.

Figure 9 shows the results of a study of the deformed metal volume directly at the crack tip for several samples of 09G2S steel. Zones with different microhardness exceeding the hardness level of undeformed metal (*180 ... 240HV0.015)* are also conventionally marked with colors on the matrices in the figure: yellow—areas with a hardness of *250...260HV0.015*; orange—*261...299HV0.015*; red—*300...330HV0.015*; maroon—above *330HV0.015*; in blue—places where hardness is not defined. The division into such ranges is nominal, the purpose of which is to demonstrate the presence of plastically deformed zones with a rough estimate of the degree of deformation.

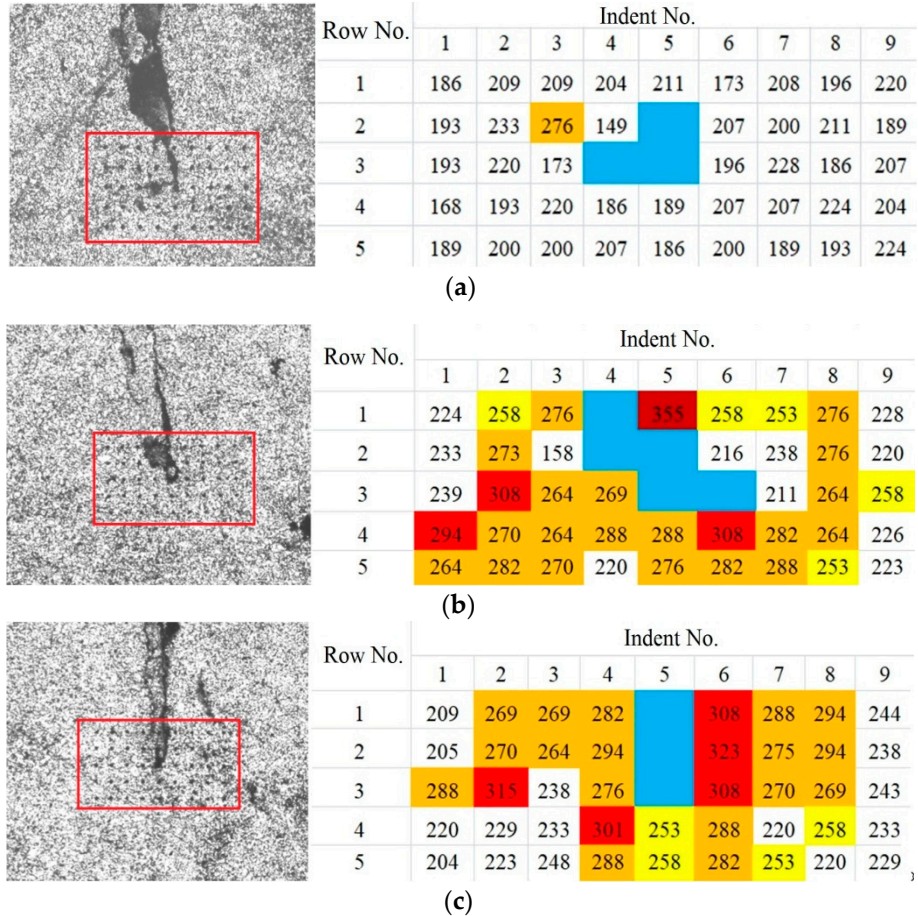

**Figure 9.** Photographs of metal near the crack tips with indents (areas of research are marked) and the *HV0.015* microhardness matrices in these areas on three samples (**a–c**) of 09G2S steel.

## 4. Discussion

This paper considered the approach to assessing the AE emissivity of structural steels during the propagation of a fatigue crack based on the Palmer–Heald model. From the point of view of evaluating the emissivity of the material, parameter $D$ is informative. Parameter $D$ characterizes the emissivity of the material and parameter $D_a$ can be used to predict the number of AE impulses $N_\Sigma$ during loading of a structure with a crack whose length is a priori unknown.

The anisotropy of the properties of the studied steels, the inhomogeneous stress–strain state of the material at the crack tip, the influence of the shape of the crack, and the presence of additional secondary sources of AE, such as friction of the crack edges, determine the random character of parameters $D$ and $D_a$ as random variables given by the law of probability distribution.

The random nature of the distribution of the parameters $D$ and $D_a$ may be due to different stress–strain states of the metal at the crack tip. Figure 9 shows that the distributions of the microhardness values (and, hence, stress and strain intensities) vary significantly from sample to sample. This is explained by the fact that in real metals and alloys, the propagation of a crack and the formation of a plastic deformation zone near its edges and apex depend not only on the absolute dimensions of the crack and loading parameters, but also on the parameters of the microstructure of the material. Among the most characteristic parameters, one should single out the grain size, the state of grain boundaries, pearlite fraction and colony size, the presence and volume fraction of non-metallic inclusions, etc. The combination of the effect of these metallurgical factors leads to differences in the shape and size of the plastic deformation zones near the crack tip.

The analysis of experimental data shows that parameters $D$ and $D_a$ can be described by the normal distribution law and this hypothesis is supported by the results of the Anderson–Darling and Pearson tests. The normal distribution is not only the most reliable but also the most conservative estimation of the distribution law, since the normal distribution corresponds to the class of "light-tailed" distribution, and its application should not lead to an overestimation of the parameter $N_\Sigma$. Using the well-known probability distribution of parameter $D_a$ based on the Palmer–Heald model, we can estimate the probability of parameter $N_\Sigma$. At a certain known level of the load $\sigma$, the probability distribution of parameter $N_\Sigma$ can be estimated by scaling the distribution of parameter $D_a$ by a $sec\left[\left(\frac{\pi}{2}\frac{\sigma}{\sigma_y}\right)-1\right]$. It is important that parameter $D_a$ does not correlate with the length of the fatigue crack—the correlation coefficient between these parameters $r_{l1,Da}$ is 0.22.

The average values of parameters $D$ and $D_a$ have minimum values for 09G2S steel, maximum values for 65G steel and intermediate values for 45 steel. 09G2S steel has a large plasticity (see Table 2 and Figure 2) compared with steels 45 and 65G. In this regard, an increase in the parameters characterizing the emissivity of the material with an increase in the carbon content in the steel structure can be explained by the more brittle nature of the fracture of steels 45 and 65G in comparison with low-carbon steel 09G2S.

It should be noted that since the amplitude discrimination threshold was set to 40 dB during the experiment, the results are quite conservative; with their help, it is possible to estimate the number of AE impulses with amplitude exceeding 40 dB. If, during the testing, the discrimination threshold is set to below 40 dB, the number of impulses recorded during AE testing will be higher. In addition, the proposed calculation model assumes that before loading, in accordance with [13], a complete load shedding was performed, otherwise, due to the Kaiser effect, the number of AE impulses will be slightly lower than expected values. It could be expected that the obtained phenomenological results concerning the random character of parameters $D$ and $D_a$ and the method of emissivity estimation, will be summarized on the class of structure steels, but the quantitative results are correct for steels of a pearlitic structural class and for thickness values of the same order that were considered above.

Comparing the present study with the studies included in the literature review, some difference in aims and approaches should be noted. At this stage, the authors did not set themselves the fundamental task of identifying the processes that occur during the propagation of a fatigue crack. The emissivity of the material is evaluated integrally using a resonant sensor recommended for industrial applications. It is assumed that the estimate of parameter $N_\Sigma$ obtained using the Palmer–Heald model can be improved for the case when the sensor is placed far from the defect on a distance characteristic of industrial testing.

## 5. Conclusions

When analyzing the data, it was found that the number of AE impulses $N_\Sigma$ emitted during loading of a sample with a fatigue crack is a random variable corresponding to the normal distribution law. The random nature of parameter $N_\Sigma$ is explained by the inhomogeneous structure of the metal at the crack tip, which leads to differences in the shape and size of the plastic deformation zones near the crack tip. An estimate of the number of AE impulses $N_\Sigma$ emitted during tension of specimens with fatigue cracks can be obtained using the distribution of the multiplicative parameter $D$ of the Palmer–Heald model, taking into account the maximum value of the applied load.

**Author Contributions:** Conceptualization, S.E.; methodology, V.B., A.M.; validation, V.B., A.M.; formal analysis, V.B.; investigation, V.B., A.M.; resources, V.B., A.M., S.E.; data curation, V.B.; writing—original draft preparation, V.B., A.M.; writing—review and editing, S.E.; visualization, V.A., A.M.; supervision, V.B., S.E.; project administration, S.E.

**Funding:** The work was carried out in the implementation of State Order Project of Ministry of Education and Science of Russian Federation in the field of scientific activity № 11.9879.2017/8.9.

**Conflicts of Interest:** The authors declare no conflict of interest.

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
