# Peer review of "Estimation of Fatigue Crack AE Emissivity Based on the Palmer–Heald Model"

_applsci, doi:10.3390/app9224851_

Round 1

Reviewer 1 Report

The paper deals with the application of the acoustic emission method to assess the parameter of the Palmer-Heald model i crack propagation.

Some integration could improve the readability.

A nomenclature is necessary.

page 2, line 86: Consider that some paper were also written to correlate the number of hits in AE and the energy amount with the fatigue phenomenon. The reliability of AE measurements (also in comparison with other NDT measurements as, for instance, Thermography) is still in progress. See, for instance, the works of  Giudice et al.: Comparison between thermal energy and acoustic emission for the fatigue behavior of steels. Structural Integrity Procedia, 18, 886-890.

page 3, line 100: I believe that a stress-strain graph would be more explicative.

page 3, line 113: Please, verify if the dimensions in Figure 1 are correct and consistent.

page 4, line 130: a table with the values of HV is necessary.

page 4, line 120: σT or σY?

In general, there is some confusion between D and Da in the text, please, verify. Is Da well defined before?

page 5, lines 188-190: please, clarify the dependence from the thickness.

page 5, line 187: the parameter N has to be better defined.

page 9, line 234: .... crack type. Figure 7 shows ...

The paper defines the normal distribution as the more reliable to model the phenomenon. Some more information about the normal parameters is necessary (average, variance, ...) for the different tests.

Author Response

Dear reviewer!

First of all, we want to thank you for the thorough analysis of our paper “Estimation of Fatigue crack AE emissivity based on Palmer-Heald Model”.

In your review, you expressed a number of comments and suggestions that will help to improve the readability of the paper. We took into account most of your wishes, and made the appropriate changes to the paper. Edited text is marked in yellow, new additions are marked in blue.

Also, in the attached file we would like to give additional comments on your suggestions.

Authors team – Vera Barat, Artem Marchenkov, Sergey Elizarov.

Reviewer 2 Report

The article requires substantial revision and cannot be published in its current form.
1. The literature review should be substantially revised and supplemented by modern literature on the issue of studying the propagation of cracks by the AE method.
In its current form, the literary review does not at all reflect the current state of affairs in the studied area. Most literary sources are over 20 years old.
2. The materials and methods section as a whole is quite detailed and understandable.
3. Sections with discussion of results and conclusions are too scarce.
The structure of the studied steels is not given. The authors should familiarize themselves with the work of Professor Vinogradov, who studied the influence of the structural state of the material on AE signals recorded during various types of deformation, including fracture.
Fractography results are not shown, although authors in Lines 245-248 cite SEM results.
Hardness measurement is an outdated and not the most reliable method for analyzing the stress-strain state of a material.

Author Response

(The authors gave the same response as above.)

Reviewer 3 Report

Dear authors,

(page 6) In Figure 4, cumulative AE hits vs. relative stress level or load were plotted together with approximation curves. From the curves, the power exponents were determined. Why didn’t you use log-log plot? The log-log plot will clearly show whether the materials conform to the Dunegan model or not. Furthermore, the power exponent can be determined more accurately. (page 7) In Figure 6, the exponents vs crack length were plotted from three materials. But only one symbol was used regardless of materials. I recommend to use three different symbols for each material. (page 6) Please explain Da more in detail. Explain how the two different distributions resulted in Figute 5. Some non-standard descriptions were found. Those are 3...5 m in line 99, and σT in line 120.

Author Response

(The authors gave the same response as above.)

Reviewer 4 Report

The submitted manuscript investigates an approach to evaluate the quantity of AE impulses emitted of fatigue crack. The methodology is based on Palmer-Heald model. The paper is well-written and the results are interesting for the engineering community. Therefore, I believe it meets the merits of the journal of Applied Sciences, and I recommend for it ‘s publication.

Author Response

Dear reviewer!

We are very grateful to you for supporting our work. Thank you!

Authors team – Vera Barat, Artem Marchenkov, Sergey Elizarov.

Round 2

Reviewer 2 Report

The manuscript has been substantially revised and may be published in its current form.

Reviewer 3 Report

Thank you for responding well to all my reviews.